# The Role of Vitamin D Supplementation in Enhancing Muscle Strength Post-Surgery: A Systemic Review

**DOI:** 10.3390/nu17091512

**Published:** 2025-04-29

**Authors:** James Jia-Dong Wang, Glenys Shu-Wei Quak, Hui-Bing Lee, Li-Xin Foo, Phoebe Tay, Shi-Min Mah, Cherie Tong, Frederick Hong-Xiang Koh

**Affiliations:** 1Lee Kong Chian School of Medicine, Nanyang Technological University, Singapore 639798, Singapore; jwang082@e.ntu.edu.sg (J.J.-D.W.); m210107@e.ntu.edu.sg (G.S.-W.Q.); 2Department of Dietetics, Sengkang General Hospital, Singapore 544886, Singapore; lee.hui.bing@skh.com.sg (H.-B.L.); phoebe.tay.d.w@skh.com.sg (P.T.); cherie.tong.c.y@skh.com.sg (C.T.); 3SingHealth Duke-NUS Muscle Health Programme, SingHealth, Singapore 168582, Singapore; foo.li.xin@skh.com.sg (L.-X.F.); mah.shi.min@skh.com.sg (S.-M.M.); 4Department of Physiotherapy, Sengkang General Hospital, Singapore 544886, Singapore; 5Department of Colorectal Surgery, Division of General Surgery, Sengkang General Hospital, Singapore 544886, Singapore; 6St Mark’s Hospital and Academic Institute, London NW10 7NS, UK; 7Yong Loo Ling School of Medicine, National University of Singapore, Singapore 119077, Singapore

**Keywords:** vitamin D, muscle strength, postoperative care, orthopaedic surgery, general surgery, supplementation

## Abstract

**Background**: Vitamin D is vital for musculoskeletal health, with emerging evidence highlighting its role in muscle function. While its preoperative and postoperative benefits for bone health are well documented, the effect of vitamin D supplementation on post-surgical muscle recovery remains underexplored. This systematic review consolidates current evidence on the impact of vitamin D supplementation in enhancing muscle strength following surgery. **Methods**: This review adhered to PRISMA guidelines and was registered on PROSPERO. A systematic search of PubMed, EMBASE, and Cochrane databases was conducted, covering articles from inception to 15 January 2025. Studies evaluating the effect of vitamin D supplementation on muscle strength in surgical contexts were included. Data extraction focused on study design, population demographics, vitamin D dosage, timing, and measured outcomes. A narrative synthesis was performed due to heterogeneity in study designs and outcomes. **Results**: From 701 initial records, 10 studies met the inclusion criteria. The findings indicate that vitamin D supplementation, particularly high-dose regimens administered preoperatively or early postoperatively, significantly improves muscle strength and functional outcomes in orthopaedic surgeries, such as hip and knee replacements, and bariatric surgeries. The benefits varied by surgical type, baseline vitamin D levels, and supplementation strategy. However, inconsistent dosing regimens and limited long-term follow-up studies hinder conclusive evidence. **Conclusions**: Vitamin D supplementation demonstrates potential in enhancing post-surgical muscle recovery and functional outcomes. Tailored supplementation strategies, based on patient-specific needs and surgical context, are essential. Future research should address optimal dosing regimens and evaluate long-term impacts on recovery and quality of life.

## 1. Introduction

Vitamin D plays an essential role in musculoskeletal health, particularly in the regulation of calcium homeostasis and bone metabolism [1]. Beyond its well-established function in bone health, increasing evidence suggests that vitamin D is crucial for muscle function as well [2]. It influences muscle strength and performance through various mechanisms, including the regulation of muscle cell metabolism, calcium influx, and muscle fibre contractility [3]. As a result, adequate vitamin D levels are associated with better muscle strength, balance, and mobility, particularly in older adults [4].

Globally, over 300 million major surgical procedures are performed annually, with this number being projected to increase in tandem with aging populations and the rising burden of chronic disease [5]. The economic cost of surgery-related care is substantial, with postoperative complications such as muscle weakness and delayed recovery contributing to longer hospital stays, readmissions, and increased healthcare expenditures [6]. For example, in the United States alone, the direct costs of surgery and related post-operative care are estimated to exceed USD 500 billion per year [7]. Enhancing postoperative recovery is therefore not only clinically significant but also economically imperative, particularly as healthcare systems shift toward value-based care models [8]. Optimising modifiable factors such as nutrition—including vitamin D status—represents a promising and cost-effective strategy to improve functional recovery and reduce complications across diverse surgical settings [9].

Preoperatively, skeletal muscle plays a vital role as a reservoir for glucose and proteins, serving as a cornerstone for metabolic stability [10]. Post-surgery, the body undergoes distinct inflammatory phases, including catabolic and anabolic stages, both of which heavily rely on muscle health [11]. During the catabolic stage, muscle proteins are broken down to supply amino acids for critical processes such as gluconeogenesis and immune system support, ensuring the body can meet the increased metabolic demands of surgical stress [12]. In the subsequent anabolic phase, adequate muscle reserves become essential for tissue repair and recovery, as they provide the building blocks required for healing and restoring physiological function [13]. Therefore, maintaining optimal muscle health preoperatively is crucial, as it not only enhances the body’s ability to buffer metabolic imbalances but also supports a smoother recovery and improved outcomes in the face of surgical stress.

The recovery of muscle strength post-surgery is also a critical determinant of functional outcomes and quality of life [14]. However, postoperative muscle weakness, often linked to prolonged immobility or insufficient physical activity, is a common challenge faced by patients after major surgeries. This is especially pronounced in procedures involving the musculoskeletal system, such as orthopaedic surgeries, or those requiring extended periods of bed rest, such as abdominal or cardiovascular operations [15]. The ability to regain muscle strength is therefore essential for reducing the risk of complications, such as falls, fractures, and prolonged hospital stays, as well as enhancing the overall rehabilitation process [16]. Optimising post-surgical muscle recovery is thus a key focus of post-operative care, as it impacts both short-term recovery and long-term physical independence.

Despite the established role of vitamin D in muscle health, the specific impact of vitamin D supplementation on post-surgical muscle recovery remains unclear [17]. While studies have shown that vitamin D status can influence post-surgical outcomes such as wound healing and infection rates in orthopaedic surgeries, its direct impact on muscle recovery following surgery remains rather underexplored [18,19]. Moreover, there is limited research that comprehensively addresses the intersection of vitamin D, post-surgical recovery beyond orthopaedic surgeries, and long-term functional outcomes [20]. This study aims to consolidate existing evidence on the impact of vitamin D supplementation on muscle strength recovery following surgery (beyond orthopaedics), with the goal of identifying key areas for future research to enhance post-operative care.

## 2. Methods

This systematic review adhered to the reporting guidelines of Preferred Reporting Items for Systematic Reviews and Meta-analyses (PRISMA) (Appendix A). It was registered with PROSPERO (International Prospective Register of Systematic Reviews; CRD42025638929).

### 2.1. Information Source and Search Strategy

A systematic search was conducted in PubMed, EMBASE, and Cochrane using Medical Subject Headings (MeSH) and keywords. Keywords and MeSH terms synonymous with “Vitamin D”, “Ergocalciferols”, “Ergosterol”, “Cholecalciferol”, “Vitamin D Deficiency”, “Surgery”, “General Surgery”, and “Muscle strength” formed the basis of the search strategy. The search period includes articles from inception to 15 January 2025. Additionally, we reviewed the reference lists of included articles and any prior systematic reviews to identify additional studies and attempted to contact authors for missing information, where appropriate. Only full-text articles published in the English language were included. The full search strategy and search terms are included in Appendix A. References were imported into Covidence to remove duplicates.

### 2.2. Study Selection

Two authors (GQSW and JDJW) reviewed each reference in a blinded manner, and any disagreements were resolved by a third independent author for the final decision (FHXK). The review was carried out in two stages: first, the titles and abstracts were reviewed, and second, the full texts of selected references were retrieved and reviewed.

Original studies, published in English, discussing vitamin D prophylaxis for muscle strength in general surgery were included. Accepted study designs included case–control, cross-sectional, cohort studies, and randomised controlled trials. Non-peer-reviewed articles, review articles (including other systematic reviews and meta-analyses), editorials, letters to editors, conference abstracts, and studies involving animal or non-human subjects were excluded.

### 2.3. Data Extraction

Two investigators (GQSW and JDJW) independently extracted information from the included studies. The data collected included authors, year of publication, type of study, country in which the study was conducted, types of surgery, dosing and timing of Vit D, number of participants, baseline demographic population characteristics (such as gender and age), outcomes that included pre- and post-operative Vit D levels, and any parameters measuring muscle strength. Where necessary, authors were contacted to retrieve missing information for clarification or completion. Discrepancies in the data extraction were resolved by consulting with a third author (FHXK).

### 2.4. Quality Assessment

The Joanna Briggs Institute Critical Appraisal Tools was used for the quality assessment of the included articles. Two investigators (GQSW and JDJW) independently reviewed all included studies, and in the case of disagreements, a third independent author (FHXK) was consulted, and a consensus was reached through discussion. The maximum score attainable (signifying high quality) was 11 points for cohort studies and 13 points for randomised controlled trials.

### 2.5. Data Analysis

The compiled data were descriptively synthesised due to the heterogeneity of the studies. Studies were further grouped by the dosage and timing of administration of vitamin D and differential effects by type of surgery (e.g., orthopaedic, spinal, transplant) for further discussion.

## 3. Results

The initial search strategy identified 701 articles, with 620 remaining after the removal of duplicates. Following title and abstract screening, 557 articles were excluded due to irrelevant title or abstract content, leaving 63 for full-text review. Of these, 53 were excluded, and a total of 10 articles were included in the final analysis (Figure 1). The study characteristics and risk of bias assessments are as described in Table 1. Although only 10 studies met the inclusion criteria, this reflects the stringent application of PRISMA guidelines and our focus on interventional trials assessing postoperative muscle strength. While the limited number may constrain generalisability, the included studies span diverse surgical contexts and consistently report trends supporting vitamin D supplementation, offering a focused yet meaningful synthesis.

### 3.1. Role of Preoperative Vitamin D Levels

Preoperative vitamin D levels play an essential role in optimising functional and recovery outcomes after surgical procedures. Studies suggest that patients with adequate preoperative vitamin D levels are more likely to recover faster and experience fewer postoperative complications compared to those who are deficient [31]. In patients undergoing total knee arthroplasty (TKA), those with sufficient preoperative vitamin D levels had significantly better functional performance postoperatively as compared to those with vitamin D deficiencies [24]. Functional outcomes, such as a wider range of joint mobility and reduced pain scores, improved immensely within the first three months following surgery. In carpal tunnel release patients, supplementing preoperative vitamin D deficiency was associated with significantly improved hand strength and quality-of-life scores. On the other hand, in general surgeries such as gastrectomy, correcting Vit D deficiency has been shown to significantly reduce declines in muscle strength and fat-free mass shortly after surgery [29]. Therefore, ensuring optimal preoperative vitamin D levels is essential for enhancing recovery and minimising complications following surgery [26].

### 3.2. Dosage and Timing of Supplementation

The efficacy of vitamin D supplementation in enhancing muscle strength post-surgery is tied to the specific dosage and timing of administration. Studies suggest that higher doses, particularly those given preoperatively or as early as possible post-surgery, can significantly improve outcomes. For instance, in patients undergoing total joint arthroplasty, a single preoperative loading dose of 300,000 IU of vitamin D3 administered within two weeks before surgery resulted in a reduction in postoperative complications, like superficial wound infections and cellulitis [30]. This strategy appears to support faster and more effective recovery by addressing potential deficiencies before the physiological stress of surgery, therefore leading to better muscle strength recovery post-surgery. In hip fracture patients, daily supplementation with 800 IU of vitamin D3 combined with calcium has been shown to improve functional recovery when initiated early post-surgery [28]. Specifically, combining a standard daily dose (800 IU) with a higher monthly dose (50,000 IU) was particularly beneficial in maintaining muscle strength following bariatric surgery, where malabsorption can complicate nutrient intake [27]. This dual-dosing regimen underscores the importance of maintaining consistent and adequate vitamin D levels, especially during the critical recovery period. Therefore, a proactive approach to supplementation—whether through preoperative loading doses or early postoperative maintenance—is essential for optimising surgical outcomes and enhancing muscle strength during recovery. Nonetheless, the duration of efficacy of high-dose vitamin D in the post-operative setting remains unclear, particularly in the context of significant muscle turnover. Furthermore, the absence of data on the half-life of vitamin D in such scenarios limits our understanding of its sustained availability and potential therapeutic benefits during the recovery period.

### 3.3. Differential Effects by Type of Surgery

The benefits of vitamin D supplementation in enhancing muscle strength and recovery outcomes vary depending on the type of surgery performed. In orthopaedic procedures, like hip fractures and total knee arthroplasty (TKA), vitamin D plays a pivotal role in functional recovery. Studies indicate that vitamin D-deficient patients undergoing TKA experience poorer functional performance and higher complication rates if the deficiency is not corrected [24]. However, when supplementation is provided postoperatively, these patients often achieve recovery outcomes comparable to those with sufficient preoperative vitamin D levels within a few months.

For hip fracture patients, combining vitamin D supplementation with calcium has shown positive effects on muscle mass, bone mineral density, and overall functional performance [21]. Specifically, in trials where patients received 800 IU to 2000 IU of vitamin D daily, improvements in lower extremity function, exemplified by faster Timed Up and Go (TUG) test times and greater knee flexor strength, were observed [28]. This indicates that even standard doses, when combined with structured physical therapy, can enhance recovery and reduce the risk of complications like muscle weakness and recurrent falls [25].

In contrast, spinal surgeries present a more complex relationship between vitamin D levels and recovery outcomes. Hypovitaminosis D is highly prevalent in patients undergoing spinal fusion and has been associated with delayed bone healing, decreased muscle strength, and increased rates of fusion failure [22]. Correcting vitamin D deficiency preoperatively can mitigate these risks by promoting bone mineralisation and supporting musculoskeletal function during recovery. However, the extent of the benefit may depend on the severity of the deficiency and the specific spinal pathology being addressed.

Vitamin D supplementation also plays a significant role in preserving muscle strength following sleeve gastrectomy (SG), a common bariatric procedure. After SG, patients often experience a decline in muscle strength due to reductions in fat-free mass (FFM), even when protein supplementation is provided. However, recent studies have shown that combining vitamin D with branched-chain amino acids (BCAAs) and whey protein can more effectively mitigate this decline compared to using whey protein alone. In a comparative analysis, patients who received a combination of whey protein, BCAAs, and 2000 IU of vitamin D exhibited a significantly lower decrease in muscle strength, with only a 3.8% reduction compared to an 18.5% reduction in those receiving protein alone [29]. This indicates that the inclusion of vitamin D plays a pivotal role in maintaining muscle function during the critical first month after surgery.

Vitamin D supports muscle strength through its interaction with receptors in muscle tissue, where it enhances calcium absorption and promotes protein synthesis [2]. These mechanisms help in maintaining muscle integrity, even during periods of rapid weight loss. Furthermore, vitamin D supplementation, in conjunction with adequate protein intake, has been associated with the preservation of handgrip strength and overall muscle function, both of which are essential for physical recovery and quality of life postoperatively [32].

These findings suggest that while orthopaedic and gastric surgeries benefit significantly from targeted vitamin D supplementation, the outcomes for spinal surgeries are influenced by a combination of factors, including baseline vitamin D status, the complexity of the surgery, and overall patient health. Therefore, personalised approaches to supplementation, tailored to the specific surgical context and patient needs, are crucial for optimising recovery and enhancing muscle strength post-surgery.

## 4. Discussion

Vitamin D is essential for several physiological processes critical to surgical recovery, including muscle strength and bone healing [33]. The studies identified highlights that patients with vitamin D deficiencies are at greater risk of poor postoperative outcomes, such as delayed recovery, increased complications, and impaired functional performance. In orthopaedic surgeries such as TKA and hip fractures, maintaining and improving muscle strength is essential for functional rehabilitation. Patients with adequate vitamin D levels exhibit faster recovery times, with improvements in mobility, balance, and strength, all of which are critical components of post-surgery rehabilitation. In contrast, vitamin D deficiency is associated with increased muscle weakness, delayed functional recovery, and a higher likelihood of recurrent falls, particularly in older adults. This highlights the need for early intervention through vitamin D supplementation to support rehabilitation goals. In bariatric surgery patients, where muscle mass and strength often decline due to rapid weight loss, vitamin D supplementation becomes especially important. Studies indicate that combining vitamin D with branched-chain amino acids (BCAAs) and whey protein can better preserve fat-free mass and handgrip strength, both of which are critical for maintaining independence and overall physical function. This illustrates that nutrition-focused interventions, including vitamin D, should be a key part of the rehabilitation process following bariatric surgery.

### 4.1. Mechanisms Linking Vitamin D to Muscle Strength

Vitamin D regulates calcium homeostasis and protein synthesis through its interaction with vitamin D receptors (VDR) expressed on skeletal muscle cells. The activation of VDR enhances muscle cell proliferation and differentiation, promoting the repair and regeneration of muscle tissue following damage or stress [34]. Studies indicate that vitamin D contributes to mitochondrial health in muscle cells, improving energy production and reducing oxidative stress, which are vital for maintaining muscle function [35]. Supplementation with vitamin D has been associated with improvements in muscle power and strength, especially in populations with baseline deficiencies [36]. Additionally, vitamin D’s ability to enhance calcium absorption supports neuromuscular function, improving muscle contraction and coordination [37]. Vitamin D also plays a role in reducing inflammation and mitigating sarcopenia by preserving lean muscle mass and preventing muscle atrophy, particularly in older adults. However, inconsistencies in findings suggest that the benefits of vitamin D on muscle strength may depend on other factors such as baseline levels and the presence of deficiencies [38]. While more research is needed to establish optimal dosing strategies, the evidence supports the role of vitamin D as a key nutrient for muscle maintenance and recovery across various populations [4].

### 4.2. Considerations of Obesity and Vitamin D Bioavailability

An important but often overlooked consideration in vitamin D supplementation is the impact of obesity and body composition on vitamin D bioavailability and efficacy [39]. Adipose tissue has been shown to sequester vitamin D, reducing its circulating bioactive levels despite supplementation [40]. This phenomenon is particularly relevant in patients with higher fat mass or sarcopenic obesity, where vitamin D may be less bioavailable despite standard dosing protocols [41]. Consequently, relying solely on serum 25(OH)D levels or BMI may be insufficient to assess true vitamin D sufficiency or predict its functional effects [42]. The assessment of vitamin D status should therefore incorporate considerations of body composition, possibly through tools such as dual-energy X-ray absorptiometry (DEXA) or bioelectrical impedance analysis, to better inform personalised dosing strategies [43]. Without adjusting for adiposity, supplementation trials may underestimate the necessary dosages required to elicit physiological benefits in muscle recovery [44]. Future studies should stratify participants by fat mass or fat-free mass and explore weight-adjusted or tissue-bioavailability-guided dosing regimens to optimise outcomes, particularly in bariatric or metabolically at-risk surgical populations [45].

### 4.3. Interplay Between Vitamin D and Other Rehabilitation Factors

While vitamin D plays a central role in recovery, it is not a standalone factor. Its effects are amplified when combined with other elements of rehabilitation, such as nutrition, physical therapy, and patient-specific care plans. For instance, vitamin D supplementation should be paired with adequate protein intake to support muscle repair and regeneration. In bariatric surgery patients, for example, the combined use of vitamin D and protein supplementation (such as whey protein and BCAAs) has been shown to reduce muscle loss, which is critical for functional recovery [46]. Additionally, physical therapy is an integral part of recovery following many surgical procedures. Vitamin D’s impact on muscle function means that it can support the rehabilitation process by allowing patients to engage more effectively in exercise and rehabilitation protocols [47]. For instance, patients with adequate vitamin D levels are more likely to respond positively to physical therapy, showing greater improvements in mobility, strength, and balance [48]. This can reduce the time spent in rehabilitation and improve long-term functional outcomes. In patients recovering from hip fractures, vitamin D supplementation, when combined with structured physical therapy, significantly improves lower extremity function and knee flexor strength, both of which are critical for regaining independence and preventing falls [28]. In these contexts, vitamin D supplementation acts as a complementary intervention, supporting the muscle strength and bone healing required for successful rehabilitation.

### 4.4. Vitamin D Status in Included Populations

Across the studies reviewed, baseline serum 25-hydroxyvitamin D [25(OH)D] levels varied widely, underscoring the heterogeneity in patient populations and potential responsiveness to supplementation. Maniar et al. reported a mean preoperative 25(OH)D level of <30 ng/mL in 53% of patients undergoing total knee arthroplasty (TKA), with significant functional improvements noted postoperatively despite deficiency [24]. In spinal surgery cohorts, Stoker et al. documented deficiency (<20 ng/mL) in 27% and insufficiency (<30 ng/mL) in 57% of adults undergoing spinal fusion, linking low levels to increased BMI and disability scores [22]. Among hip fracture patients, Stemmle et al. observed baseline deficiency in a large proportion, although the mean values were not always explicitly stated; their findings indicated a positive response to 800 IU of vitamin D3 when combined with exercise, suggesting benefit even at modest repletion levels [28]. In bariatric populations, Schiavo et al. showed that even patients with 25(OH)D levels above 75 nmol/L experienced muscle strength decline, albeit attenuated with combined supplementation [29]. These data reinforce the importance of assessing vitamin D status not merely by absolute thresholds but in context with body composition, surgical type, and co-interventions. Future studies should standardise the reporting of baseline and post-intervention 25(OH)D levels to allow for more nuanced comparisons and dosing recommendations.

### 4.5. Clinical Implications

The clinical implications of these findings are substantial. Given the widespread prevalence of vitamin D deficiency, particularly in populations undergoing surgery, there is a clear need for routine screening and supplementation strategies aimed at correcting vitamin D deficiencies before surgery and throughout the recovery period [20]. Additionally, vitamin D supplementation should not be seen as a one-size-fits-all solution but rather as a tailored intervention based on the patient’s baseline vitamin D levels, the type of surgery, and the individual’s overall health [49]. For example, patients undergoing bariatric surgery or those with malabsorptive conditions may require higher doses or a combination of supplementation strategies to counteract nutrient deficiencies that can complicate recovery [50]. Similarly, patients with spinal conditions or those at risk for fusion failure may benefit from early and sustained vitamin D supplementation to support bone healing and muscle strength [51].

### 4.6. Research Implications

The role of vitamin D supplementation in enhancing muscle strength post-surgery, while promising, warrants further investigation through more robust and large-scale trials. Although several studies have demonstrated positive outcomes in specific surgical contexts, such as hip fractures and total joint arthroplasty, the variability in dosage, timing, and patient populations highlights the need for standardised, evidence-based protocols [28,30]. Future research should focus on identifying optimal dosing regimens tailored to different types of surgeries and patient profiles, including those with varying degrees of vitamin D deficiency.

Randomised controlled trials (RCTs) comparing preoperative loading doses with postoperative maintenance dosing across multiple surgical specialties could provide clarity on the most effective strategies for improving functional outcomes. Additionally, integrating vitamin D supplementation with physical rehabilitation programs—such as home exercise regimens or physical therapy—may offer synergistic benefits for muscle recovery [25,28]. These combined interventions warrant exploration to determine the most effective multidisciplinary approach to postoperative care. Another important consideration for future research is the inclusion of diverse patient demographics, like younger adults undergoing spinal fusions and elderly populations with comorbidities like osteoporosis or obesity. This diversity can help tailor supplementation protocols to specific clinical needs.

Moreover, current evidence is limited by inconsistencies in study design, small sample sizes, and heterogeneous patient populations. To address these gaps, future trials should adopt standardised methodologies, including uniform criteria for vitamin D deficiency, consistent dosing schedules, and clearly defined endpoints for functional recovery [52]. Longitudinal studies extending beyond the typical six-to-twelve-month follow-up period are essential to assess the sustained effects of vitamin D on muscle strength and functional outcomes [22]. Furthermore, the impact of vitamin D supplementation on specific surgical complications, such as infection rates and delayed healing, remains underexplored [30]. Investigating these outcomes could provide valuable insights into the broader clinical benefits of correcting vitamin D deficiency in surgical patients.

### 4.7. Knowledge Gaps

Despite growing evidence supporting the benefits of vitamin D supplementation, there remains a lack of standardised protocols for its administration in surgical settings. The studies reviewed demonstrated wide variability in dosing regimens, ranging from daily doses of 800 IU to large preoperative boluses of 300,000 IU [28,30]. This inconsistency makes it challenging to develop clear guidelines for clinicians.

Establishing a consensus on optimal dosage, frequency, and timing—both pre- and post-surgery—is crucial for ensuring effective implementation and maximising patient outcomes. Additionally, data on long-term functional outcomes remain limited. Most studies focus on short-to-medium-term recovery periods of six to twelve months [24,25]. However, the potential for vitamin D to influence long-term muscle strength, mobility, and quality of life is not well understood. Given that muscle weakness and bone loss can persist for years after surgery, particularly in elderly or frail patients, extended follow-up studies are necessary to capture the full spectrum of recovery [26].

Moreover, while evidence exists for orthopaedic surgeries, such as hip and knee replacements, fewer studies have explored the role of vitamin D in spinal surgeries and other less common procedures [22]. Research focusing on these areas could address important gaps and broaden the applicability of supplementation protocols across various surgical fields.

Beyond reiterating current evidence, this review uniquely highlights the importance of timing, dosage, and surgical context in optimising vitamin D supplementation for postoperative muscle recovery—a nuance often overlooked in the prior literature. By synthesising findings across orthopaedic, spinal, and bariatric surgeries, we underscore the need for tailored protocols that consider both the patient’s baseline status and the nature of the surgery. Clinically, our review supports the implementation of perioperative vitamin D screening and stratified supplementation regimens as feasible, low-risk strategies to enhance rehabilitation and reduce complications. This offers actionable insights for surgeons, dietitians, and physiotherapists aiming to improve functional outcomes and long-term quality of life in surgical patients. As healthcare systems increasingly prioritise value-based care, integrating evidence-based nutritional interventions like vitamin D supplementation into surgical recovery pathways represents a practical, cost-effective approach to optimising patient outcomes.

Future research should build on these knowledge gaps by addressing the variability in current vitamin D supplementation practices and expanding the scope of investigation. Studies like the D-ECISIVE trial (NCT06708741) would provide valuable insights but highlight critical limitations, such as the need for standardised dosing protocols that balance safety and efficacy across diverse surgical populations. Investigating the interplay between vitamin D supplementation and other perioperative interventions, such as nutritional support or physical rehabilitation, could provide a more holistic understanding of its role in recovery [53]. Additionally, incorporating biomarkers like inflammatory markers and genetic predispositions affecting vitamin D metabolism may uncover mechanisms underlying its effects on muscle recovery [54]. Multi-centre, randomised trials with diverse demographic cohorts and extended follow-up periods beyond 12 months are essential to evaluate the long-term impacts of supplementation on muscle strength, mobility, and quality of life [31]. Expanding research to include underrepresented surgical areas, such as spinal and oncological procedures, will further refine and broaden the applicability of vitamin D supplementation protocols, ensuring they are tailored to a variety of clinical contexts [55].

### 4.8. Strengths and Limitations

This review has several strengths. It is one of the few to synthesise evidence across a broad spectrum of surgical contexts, integrating data from orthopaedic, bariatric, and spinal procedures to draw context-specific conclusions about vitamin D supplementation and muscle strength recovery.

However, several limitations warrant consideration. First, the heterogeneity in study designs, dosing regimens, and outcome measures limited the feasibility of meta-analysis, and a narrative synthesis was used instead. Second, the majority of included studies were conducted in orthopaedic populations, with fewer high-quality trials being available in other surgical domains, potentially skewing generalisability. Third, the variable definitions of vitamin D deficiency and inconsistencies in reporting baseline levels posed challenges in drawing definitive conclusions regarding optimal dosing thresholds. Lastly, publication bias cannot be excluded, as studies with negative or null findings may be underrepresented in the literature. Despite these limitations, this review provides a timely and practical synthesis that can inform perioperative care and guide future research directions.

## 5. Conclusions

In conclusion, while vitamin D supplementation shows potential for enhancing muscle strength and improving functional recovery post-surgery, significant gaps in evidence remain. Future research should prioritise standardised protocols, diverse patient populations, and long-term follow-up to fully elucidate the benefits and optimise clinical guidelines. Addressing these gaps will not only improve surgical outcomes but also enhance the quality of life for patients recovering from surgery.

## Figures and Tables

**Figure 1 nutrients-17-01512-f001:**
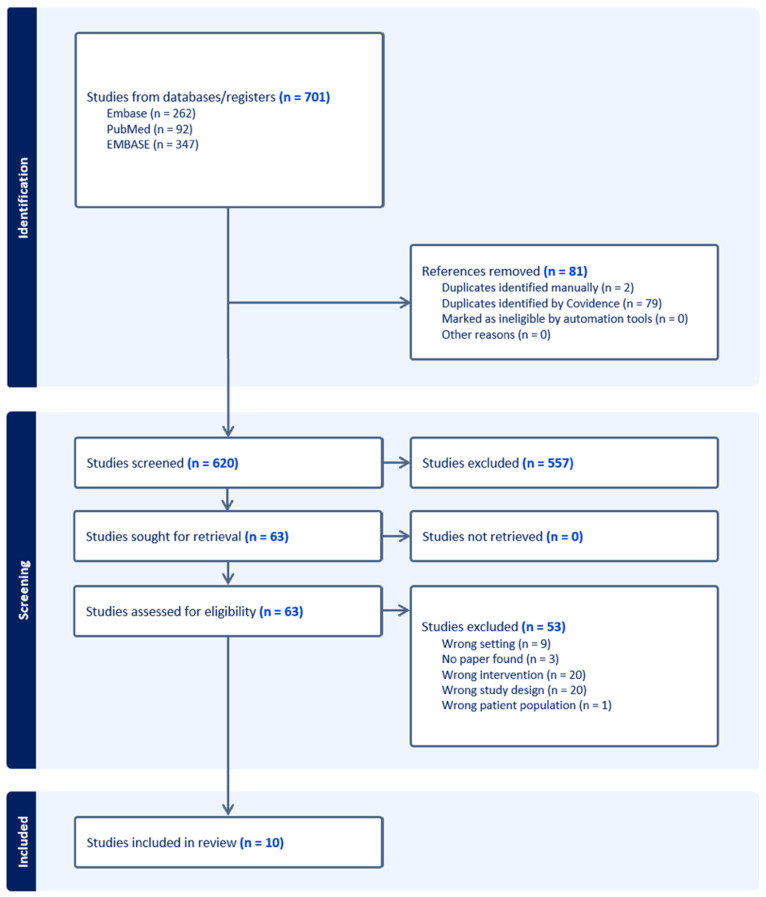
PRISMA (Preferred Reporting Items for Systematic Reviews and Meta-Analyses) flowchart for systematic reviews, including searches of databases and registers.

**Table 1 nutrients-17-01512-t001:** Summary of studies included in scoping review.

References	Title of Study	Year	Authors	Type of Study	Country	Type of Surgery	Dose of Vitamin D	Key Outcomes Measured	JBI Score
[21]	Positive Effects of Anabolic Steroids, Vitamin D, and Calcium on Muscle Mass, Bone Mineral Density, and Clinical Function After a Hip Fracture: A Randomised Study of 63 Women	2002	Hedstrom, M, Sjoberg, K, Brosjo, E, Astrom, K, Sjoberg, H, Dalen, N	Randomised Controlled Trial	Sweden	Orthopaedics, Hip Replacement	1-alpha-hydroxylated vitamin D3 (alphacalcidol 0.25 g)	**Muscle volume**: The anabolic group maintained muscle volume; the control group experienced loss (*p* < 0.01). **Bone density**: Less bone loss in the anabolic group. **Clinical outcomes**: Improved Harris hip scores and gait speed in the anabolic group.	**11/13**
[22]	Preoperative Vitamin D Status in Adults Undergoing Spinal Fusion	2011	Buchowski J., Stoker G., Bridwell K., Lenke L., Daniel Riew K.K., Zebala L.	Cohort Study	United States	Orthopaedics, Spinal Fusion	Vitamin D deficiency defined as <20 ng/mL; supplementation regimen: 50,000 IU D2 (ergocalciferol)	**Deficiency prevalence**: 57% had inadequate levels, 27% deficient. **Associated factors**: Higher BMI, smoking, lack of supplementation linked to deficiency.	**10/11**
[23]	Effects of Protein-Rich Nutritional Supplementation and Bisphosphonates on Body Composition, Handgrip Strength, and Quality of Life After Hip Fracture: A 12-month Randomised Controlled Study	2015	Flodin L, Cederholm T, Sääf M, Samnegård E, Ekström W, Al-Ani AN, Hedström M	Randomised Controlled Trial	Sweden	Orthopaedics, Hip Replacement	800 IU daily vitamin D3 for all study groups	**Handgrip strength (HGS)**: Improvement between baseline and six months in the nutritional supplementation group (*p* = 0.04). **Health-related quality of life (HRQoL)**: Decreased in controls but stable in the nutritional group.	**11/13**
[24]	Effect of Preoperative Vitamin D Levels on Functional Performance After Total Knee Arthroplasty	2016	Maniar RN, Patil AM, Maniar AR, Gangaraju B, Singh J	Cohort Study	India	Orthopaedics, Knee Replacement	Vitamin D supplementation postoperatively (0.5 µg/day)	**Pre-surgery function**: Patients with vitamin D deficiency had lower preoperative WOMAC scores (*p* = 0.040). **Postoperative recovery**: Vitamin D supplementation post-surgery normalized outcomes within three months.	**10/11**
[25]	Timeline of Functional Recovery After Hip Fracture in Seniors Aged 65 and Older: A Prospective Observational Analysis	2019	Fischer K, Trombik M, Freystätter G, Egli A, Theiler R, Bischoff-Ferrari HA	Randomised Controlled Trial	Switzerland	Orthopaedics, Hip Replacement	800 IU or 2000 IU daily of vitamin D3 (from primary study protocol)	**Recovery milestones**: Objective improvements in function (TUG, +61%) occurred within six months, while subjective recovery (SF-36) extended to nine months. **Grip strength**: Continued to decline at both 6 and 12 months.	**10/11**
[26]	The Effect of Vitamin D Deficiency Correction on the Outcomes in Women After Carpal Tunnel Release	2019	Lee MH, Gong HS, Cho KJ, Kim J, Baek GH	Cohort Study	South Korea	Orthopaedics, Carpal Tunnel Release	1000 IU daily vitamin D3 for six months	**Functional improvement**: Patients who corrected deficiency had better DASH scores. **Strength and nerve velocity**: No correlation between vitamin D levels and grip/pinch strength or motor conduction velocity.	**10/11**
[27]	Assessment of Physical Fitness After Bariatric Surgery and Its Association With Protein Intake and Type of Cholecalciferol Supplementation	2019	Smelt HJM, Pouwels S, Celik A, Gupta A, Smulders JF	Cohort Study	Netherlands	General Surgery, Bariatric Surgery	Group A: 800 IU daily; Group B: 800 IU daily + 50,000 IU monthly vitamin D3	**Handgrip strength (HS)**: Protein intake significantly influenced HS (*p* = 0.017). **Shuttle Walk Run Test (SWRT)**: No significant impact of vitamin D levels or supplementation observed.	**10/11**
[28]	Effect of 800 IU Versus 2000 IU Vitamin D3 With or Without a Simple Home Exercise Program on Functional Recovery After Hip Fracture: A Randomised Controlled Trial	2019	Stemmle J, Marzel A, Chocano-Bedoya PO, Orav EJ, Dawson-Hughes B, Freystaetter G, Egli A, Theiler R, Staehelin HB, Bischoff-Ferrari HA	Randomised Controlled Trial	Switzerland	Orthopaedics, Hip Replacement	800 IU or 2000 IU daily of vitamin D3	**TUG improvement**: Combining 800 IU vitamin D3 with a home exercise program resulted in significantly better TUG test scores (13.8 s vs. 19.5 s in standard care). **Knee strength**: Improvements in knee flexor strength (approached significance). **Subjective scores**: No significant improvements in subjective physical functioning scores (PF-10).	**11/13**
[29]	Adding Branched-Chain Amino Acids and Vitamin D to Whey Protein Is More Effective Than Protein Alone in Preserving Fat-Free Mass and Muscle Strength After Sleeve Gastrectomy	2024	Schiavo L, Santella B, Paolini B; Rahimi F, Giglio E, Martinelli B, Boschetti S, Bertolani L, Gennai K, Arolfo S, Bertani MP, Pilone V	Cohort Study	Italy	General Surgery, Gastrectomy	Vitamin D dose integrated into supplementation regimen (2000 IU)	**Fat-free mass (FFM) and muscle strength (MS)**: Less reduction in FFM and MS with combined supplementation (4.1% vs. 11.4% FFM; 3.8% vs. 18.5% MS). **Body weight**: No difference in total body weight loss.	**11/13**
[30]	Effect of Vitamin D Deficiency on Periprosthetic Joint Infection and Complications After Primary Total Joint Arthroplasty	2024	Birinci M., Hakyemez O.S., Geckalan M.A., Mutlu M.; Yildiz F.; Bilgen O.F., Azboy I.	Cohort Study	Turkey	Orthopaedics, Hip Or Knee Replacement	Vitamin D deficiency corrected with a single dose of 300,000 IU D3 two weeks preoperatively	**Complications**: Higher rates of wound infections and cellulitis in vitamin D-deficient patients. **PJI risk**: No significant difference in periprosthetic joint infections.	**10/11**

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
