# Peer review of "The Role of Vitamin D Supplementation in Enhancing Muscle Strength Post-Surgery: A Systemic Review"

_nutrients, 2025, doi:10.3390/nu17091512_

Round 1
Reviewer 1 Report
Comments and Suggestions for Authors
Dear Authors,
After reading your interesting research, I have some comments I would like you to address listed below:
- Many articles investigate vitamin D's role in human health. Review articles on vitamin D in orthopaedic surgeries (10.1016/j.otsr.2018.12.006 and doi.org/10.3390/nu13051675). Please clarify the novelty of your study and your contribution to the field.
2. The conclusion sounds rather obvious. Please convince the reviewers and potential readers that your research adds something to the subject area compared with other published material, instead of simply summarizing the current knowledge. Some practical implications can elevate your work.
3. Please correct minor typos throughout your manuscript.
4. The information about the search period described in the abstract and the methodology is inconsistent. Please correct.
5. The methodology is transparent and well-described.
6. The discussion is acceptable. I appreciate the emphasis on clinical implications. However, this paragraph might be improved and supplemented because it is too general.
I want to express my positive impression, which may interest the journal’s readers after some necessary corrections.
Best regards,
The reviewer.
Author Response
Dear Reviewers,
Thank you very much for the opportunity to revise our manuscript. We wish to thank the reviewers for taking the time to review our paper and provide comprehensive feedback. We greatly appreciate your insightful comments and suggestions, which will undoubtedly contribute to improving the quality and clarity of our work. Below, we outline the revisions and clarifications made in response to the feedback (in blue text)
Reviewer #1
After reading your interesting research, I have some comments I would like you to address listed below:
- Many articles investigate vitamin D's role in human health. Review articles on vitamin D in orthopaedic surgeries (10.1016/j.otsr.2018.12.006 and doi.org/10.3390/nu13051675). Please clarify the novelty of your study and your contribution to the field.
We have added in a paragraph to clarify the novelty of the study which emphasis on muscle strength and also to look beyond orthopaedic surgery.
“Despite the established role of vitamin D in muscle health, the specific impact of vitamin D supplementation on post-surgical muscle recovery remains unclear. While studies have shown that vitamin D status can influence post-surgical outcomes such as wound healing and infection rates in orthopaedic surgeries, its direct impact on muscle recovery following surgery remains rather underexplored9,10. Moreover, there is limited research that comprehensively addresses the intersection of vitamin D, post-surgical recovery beyond orthopaedic surgeries, and long-term functional outcomes11. This study aims to consolidate existing evidence on the impact of vitamin D supplementation on muscle strength recovery following surgery (beyond orthopaedics), with the goal of identifying key areas for future research to enhance post-operative care.”
- The conclusion sounds rather obvious. Please convince the reviewers and potential readers that your research adds something to the subject area compared with other published material, instead of simply summarizing the current knowledge. Some practical implications can elevate your work. We have added a paragraph on practical implications in the discussion.
“Beyond reiterating current evidence, this review uniquely highlights the importance of timing, dosage, and surgical context in optimizing vitamin D supplementation for postoperative muscle recovery—a nuance often overlooked in prior literature. By synthesizing findings across orthopedic, spinal, and bariatric surgeries, we underscore the need for tailored protocols that consider both the patient's baseline status and the nature of the surgery. Clinically, our review supports the implementation of perioperative vitamin D screening and stratified supplementation regimens as feasible, low-risk strategies to enhance rehabilitation and reduce complications. This offers actionable insights for surgeons, dietitians, and physiotherapists aiming to improve functional outcomes and long-term quality of life in surgical patients. As healthcare systems increasingly prioritize value-based care, integrating evidence-based nutritional interventions like vitamin D supplementation into surgical recovery pathways represents a practical, cost-effective approach to optimizing patient outcomes.”
- Please correct minor typos throughout your manuscript.
We have corrected the minor typos
- The information about the search period described in the abstract and the methodology is inconsistent. Please correct.
We have checked the manuscript and amended it.
“This review adhered to PRISMA guidelines and was registered on PROSPERO. A systematic search of PubMed, EMBASE, and Cochrane databases was conducted, covering articles from inception to 15 Jan, 2025.”
- The methodology is transparent and well-described.
- The discussion is acceptable. I appreciate the emphasis on clinical implications. However, this paragraph might be improved and supplemented because it is too general.
We have added in practical implications and also consideration of BMI in such patients.
“An important but often overlooked consideration in vitamin D supplementation is the impact of obesity and body composition on vitamin D bioavailability and efficacy. Adipose tissue has been shown to sequester vitamin D, reducing its circulating bioactive levels despite supplementation. This phenomenon is particularly relevant in patients with higher fat mass or sarcopenic obesity, where vitamin D may be less bioavailable despite standard dosing protocols. Consequently, relying solely on serum 25(OH)D levels or BMI may be insufficient to assess true vitamin D sufficiency or predict its functional effects. The assessment of vitamin D status should therefore incorporate considerations of body composition, possibly through tools such as dual-energy X-ray absorptiometry (DEXA) or bioelectrical impedance analysis, to better inform personalized dosing strategies. Without adjusting for adiposity, supplementation trials may underestimate the necessary dosages required to elicit physiological benefits in muscle recovery. Future studies should stratify participants by fat mass or fat-free mass and explore weight-adjusted or tissue-bioavailability–guided dosing regimens to optimize outcomes, particularly in bariatric or metabolically at-risk surgical populations.”
“Beyond reiterating current evidence, this review uniquely highlights the importance of timing, dosage, and surgical context in optimizing vitamin D supplementation for postoperative muscle recovery—a nuance often overlooked in prior literature. By synthesizing findings across orthopedic, spinal, and bariatric surgeries, we underscore the need for tailored protocols that consider both the patient's baseline status and the nature of the surgery. Clinically, our review supports the implementation of perioperative vitamin D screening and stratified supplementation regimens as feasible, low-risk strategies to enhance rehabilitation and reduce complications. This offers actionable insights for surgeons, dietitians, and physiotherapists aiming to improve functional outcomes and long-term quality of life in surgical patients. As healthcare systems increasingly prioritize value-based care, integrating evidence-based nutritional interventions like vitamin D supplementation into surgical recovery pathways represents a practical, cost-effective approach to optimizing patient outcomes.”
I want to express my positive impression, which may interest the journal’s readers after some necessary corrections.
Reviewer 2 Report
Comments and Suggestions for Authors
The subject of this article is the importance of vitamin D supplementation to increase muscle strength in patients after various types of surgeries.
In the Introduction, the authors outline the role of vitamin D, its mechanisms, and its impact on muscle strength both pre- and postoperatively.
In the Methodology section, the authors describe the search strategy, study selection, and data analysis.
The Results and Discussion sections clearly present different aspects of the role and importance of vitamin D supplementation and dosage.
*Lack of limitation of the research.
Author Response
Dear Reviewers,
Thank you very much for the opportunity to revise our manuscript. We wish to thank the reviewers for taking the time to review our paper and provide comprehensive feedback. We greatly appreciate your insightful comments and suggestions, which will undoubtedly contribute to improving the quality and clarity of our work. Below, we outline the revisions and clarifications made in response to the feedback (in blue text)
Thank you for your comments, we have added it in a limitations paragraph.
“This review has several strengths. It is one of the few to synthesize evidence across a broad spectrum of surgical contexts, integrating data from orthopedic, bariatric, and spinal procedures to draw context-specific conclusions about vitamin D supplementation and muscle strength recovery.
However, several limitations warrant consideration. First, the heterogeneity in study designs, dosing regimens, and outcome measures limited the feasibility of meta-analysis, and a narrative synthesis was used instead. Second, the majority of included studies were conducted in orthopedic populations, with fewer high-quality trials available in other surgical domains, potentially skewing generalizability. Third, the variable definitions of vitamin D deficiency and inconsistencies in reporting baseline levels posed challenges in drawing definitive conclusions regarding optimal dosing thresholds. Lastly, publication bias cannot be excluded, as studies with negative or null findings may be underrepresented in the literature. Despite these limitations, this review provides a timely and practical synthesis that can inform perioperative care and guide future research directions.”
Reviewer 3 Report
Comments and Suggestions for Authors
The reviewer would like to offer the following points for the authors' consideration:
- Consider including some statistics (in the introduction section) in terms of surgeries and cost/burden on the economy at the global level.
- The consideration of obesity (beyond BMI) but most importantly regarding body composition is important when it comes to vitamin D. Especially given that vitamin D can become less bioavailable at higher levels of obesity since it can at least in part be captured in the adipose tissue. A discussion of the topic would strengthen the paper and essentially bring up the issue of assessment of vitamin D status which is key in terms of drawing actual and meaningful association between vitamin D and outcomes. The mere supplementation may be misleading especially when we strive to produce recommendation and regimes for dosing.
- A discussion on the actual vitamin D status would be important and strengthen the paper. What was the 25(OH)D levels in those patients?
- The n of studies finally considered for the review (n=10) is rather low. While I understand that the exclusion criteria and the PRISMA process rendered this number it is worth questioning and discussing how competing the results and conclusions are when only 10 studies are considered from the vast literature.
- The manuscript is significantly undereferrenced for a review article.
Author Response
Dear Reviewers,
Thank you very much for the opportunity to revise our manuscript. We wish to thank the reviewers for taking the time to review our paper and provide comprehensive feedback. We greatly appreciate your insightful comments and suggestions, which will undoubtedly contribute to improving the quality and clarity of our work. Below, we outline the revisions and clarifications made in response to the feedback (in blue text)
Reviewer #3
The reviewer would like to offer the following points for the authors' consideration:
Consider including some statistics (in the introduction section) in terms of surgeries and cost/burden on the economy at the global level.
We have added a paragraph in the introduction to outline this,
“Globally, over 300 million major surgical procedures are performed annually, with this number projected to increase in tandem with aging populations and the rising burden of chronic disease. The economic cost of surgery-related care is substantial, with postoperative complications such as muscle weakness and delayed recovery contributing to longer hospital stays, readmissions, and increased healthcare expenditures. For example, in the United States alone, the direct costs of surgery and related post-operative care are estimated to exceed $500 billion per year. Enhancing postoperative recovery is therefore not only clinically significant but also economically imperative, particularly as healthcare systems shift toward value-based care models. Optimizing modifiable factors such as nutrition—including vitamin D status—represents a promising and cost-effective strategy to improve functional recovery and reduce complications across diverse surgical settings.”
The consideration of obesity (beyond BMI) but most importantly regarding body composition is important when it comes to vitamin D. Especially given that vitamin D can become less bioavailable at higher levels of obesity since it can at least in part be captured in the adipose tissue. A discussion of the topic would strengthen the paper and essentially bring up the issue of assessment of vitamin D status which is key in terms of drawing actual and meaningful association between vitamin D and outcomes. The mere supplementation may be misleading especially when we strive to produce recommendation and regimes for dosing.
We have added a paragraph in the discussion to outline this,
“An important but often overlooked consideration in vitamin D supplementation is the impact of obesity and body composition on vitamin D bioavailability and efficacy. Adipose tissue has been shown to sequester vitamin D, reducing its circulating bioactive levels despite supplementation. This phenomenon is particularly relevant in patients with higher fat mass or sarcopenic obesity, where vitamin D may be less bioavailable despite standard dosing protocols. Consequently, relying solely on serum 25(OH)D levels or BMI may be insufficient to assess true vitamin D sufficiency or predict its functional effects. The assessment of vitamin D status should therefore incorporate considerations of body composition, possibly through tools such as dual-energy X-ray absorptiometry (DEXA) or bioelectrical impedance analysis, to better inform personalized dosing strategies. Without adjusting for adiposity, supplementation trials may underestimate the necessary dosages required to elicit physiological benefits in muscle recovery. Future studies should stratify participants by fat mass or fat-free mass and explore weight-adjusted or tissue-bioavailability–guided dosing regimens to optimize outcomes, particularly in bariatric or metabolically at-risk surgical populations.”
A discussion on the actual vitamin D status would be important and strengthen the paper. What was the 25(OH)D levels in those patients?
We have included it and the implications within the updated discussion section.
“Across the studies reviewed, baseline serum 25-hydroxyvitamin D [25(OH)D] levels varied widely, underscoring the heterogeneity in patient populations and potential responsiveness to supplementation. Maniar et al. reported a mean preoperative 25(OH)D level of <30 ng/mL in 53% of patients undergoing total knee arthroplasty (TKA), with significant functional improvements noted postoperatively despite deficiency​. In spinal surgery cohorts, Stoker et al. documented deficiency (<20 ng/mL) in 27% and insufficiency (<30 ng/mL) in 57% of adults undergoing spinal fusion, linking low levels to increased BMI and disability scores​. Among hip fracture patients, Stemmle et al. observed baseline deficiency in a large proportion, although the mean values were not always explicitly stated; their findings indicated a positive response to 800 IU of vitamin D3 when combined with exercise, suggesting benefit even at modest repletion levels​. In bariatric populations, Schiavo et al. showed that even patients with 25(OH)D levels above 75 nmol/L experienced muscle strength decline, albeit attenuated with combined supplementation​. These data reinforce the importance of assessing vitamin D status not merely by absolute thresholds but in context with body composition, surgical type, and co-interventions. Future studies should standardize reporting of baseline and post-intervention 25(OH)D levels to allow for more nuanced comparisons and dosing recommendations.”
The n of studies finally considered for the review (n=10) is rather low. While I understand that the exclusion criteria and the PRISMA process rendered this number it is worth questioning and discussing how competing the results and conclusions are when only 10 studies are considered from the vast literature.
We have added a few lines to outline this limitations.
“Although only 10 studies met the inclusion criteria, this reflects the stringent application of PRISMA guidelines and our focus on interventional trials assessing postoperative muscle strength. While the limited number may constrain generalizability, the included studies span diverse surgical contexts and consistently report trends supporting vitamin D supplementation, offering a focused yet meaningful synthesis.”
The manuscript is significantly undereferrenced for a review article.
We have added in additional references based on the feedback from reviewers.